Identification of hub genes and small molecule therapeutic drugs related to breast cancer with comprehensive bioinformatics analysis

Hao Mingqian
Liu Wencong xiaodiliqian@163.com
Ding Chuanbo
Peng Xiaojuan
Zhang Yue
Chen Huiying
Dong Ling
Liu Xinglong
Zhao Yingchun
Chen Xueyan
Khatoon Sadia
Zheng Yinan
School of Chinese Medicinal Materials, Jilin Agricultural University , Changchun , Jilin , China
Karakülah Gökhan
Electronic publication date: 2020 Sep 29
Publication date: 2020
Volume: 8
Electronic Location ID: e9946
Received 2020 Jun 11; Accepted 2020 Aug 25
Copyright: ©2020 Hao et al.
Copyright year: 2020
Copyright holder: Hao et al.
License: This is an open access article distributed under the terms of the Creative Commons Attribution License, which permits unrestricted use, distribution, reproduction and adaptation in any medium and for any purpose provided that it is properly attributed. For attribution, the original author(s), title, publication source (PeerJ) and either DOI or URL of the article must be cited.
License URL: https://creativecommons.org/licenses/by/4.0/

Keywords: Breast cancer, Biomarkers, Traditional Chinese medicine, Bioinformatics analysis, Immune infiltration

Funding: Natural Science Foundation of Jilin Province 20191102008YY This research was funded by the Natural Science Foundation of Jilin Province (No. 20191102008YY). The funders had no role in study design, data collection and analysis, decision to publish, or preparation of the manuscript.

==============================
Breast cancer is one of the most common malignant tumors among women worldwide and has a high morbidity and mortality. This research aimed to identify hub genes and small molecule drugs for breast cancer by integrated bioinformatics analysis. After downloading multiple gene expression datasets from The Cancer Genome Atlas (TCGA) and Gene Expression Omnibus (GEO) database, 283 overlapping differentially expressed genes (DEGs) significantly enriched in different cancer-related functions and pathways were obtained using LIMMA, VennDiagram and ClusterProfiler packages of R. We then analyzed the topology of protein–protein interaction (PPI) network with overlapping DEGs and further obtained six hub genes (RRM2, CDC20, CCNB2, BUB1B, CDK1, and CCNA2) from the network via STRING and Cytoscape. Subsequently, we conducted genes expression verification, genetic alterations evaluation, immune infiltration prediction, clinicopathological parameters analysis, identification of transcriptional and post-transcriptional regulatory molecules, and survival analysis for these hub genes. Meanwhile, 29 possible drug candidates (e.g., Cladribine, Gallium nitrate, Alvocidib, 1β-hydroxyalantolactone, Berberine hydrochloride, Nitidine chloride) were identified from the DGIdb database and the GSE85871 dataset. In addition, some transcription factors and miRNAs (e.g., E2F1, PTTG1, TP53, ZBTB16, hsa-miR-130a-3p, hsa-miR-204-5p) targeting hub genes were identified as key regulators in the progression of breast cancer. In conclusion, our study identified six hub genes and 29 potential drug candidates for breast cancer. These findings may advance understanding regarding the diagnosis, prognosis and treatment of breast cancer.

Introduction

Among cancers affecting females, breast cancer has a particularly high incidence, recurrence and mortality rate. Although encouraging progress has been made in the early diagnosis and systemic treatment over several decades, the overall 5-year survival rate for patients with breast cancer is still low, and the incidence rate continues to increase annually (Bray et al., 2018; Waks & Winer, 2019). Breast cancer has become a serious public health problem all over the world, which brings great economic burdens to individuals and families. Thus, more efforts are required to find effective biomarkers for the early diagnosis, accurate prognosis, and targeted therapy of breast cancer.

High-throughput sequencing technology and bioinformatics analysis methods can reveal changes in the expression of a vast amount of genes and are also effective tools to identify candidate biomarkers for breast cancer research (Kulasingam & Diamandis, 2008). Liu et al. (2020) explored the expression and prognostic value of TDO2 in breast cancer using transcriptome data, and analyzed the correlation between TDO2 gene and tumor immune invasion, suggesting that TDO2 was a promising new immunotherapy target for breast cancer. Others (Li et al., 2020b) studied the biological mechanism of BRCA1/2 mutant breast cancer and evaluated the diagnosis and prognosis values of key genes using bioinformatics methods. In addition, to clarify the role of low-frequency mutated genes in breast cancer, Lusito et al. (2019) used functional network construction, gene mutation analysis, hierarchical clustering and cancer module recognition to analyze the gene expression and mutation datasets of breast cancer. Similarly, Wang et al. (2019) used research samples from The Cancer Genome Atlas (TCGA, https://www.cancer.gov/about-nci/organization/ccg/research/structural-genomics/tcga) and Gene Expression Omnibus (GEO, https://www.ncbi.nlm.nih.gov/geo/) database to analyze the pathogenesis and potential prognostic marker genes of breast cancer. In general, the majority of bioinformatics studies focused on how key genes affect the tumorigenesis and prognosis of breast cancer, with limited research on the systemic analysis of multi-gene and multi-pathway as well as potential drugs.

In the present study, we applied comprehensive bioinformatics methods, such as differentially expressed genes (DEGs) identification, protein–protein interaction (PPI) network integration, hub genes identification, correlation prediction between hub genes, determination of genetic alterations, Gene Ontology (GO) and Kyoto Encyclopedia of Genes and Genomes (KEGG) pathway enrichment, immune cell infiltrates evaluation, clinicopathological features analysis and survival analysis to process large-scale DNA microarrays and RNA-seq data from GEO, TCGA and other public databases so as to explore potential hub genes and biological pathways related to the occurrence, development and prognosis of breast cancer. We also explored the transcription factors (TFs) and miRNAs that regulate the transcriptional and post-transcriptional processes of hub genes, respectively. In addition, several small molecule drugs for breast cancer were obtained from the DGIdb drug relocation database (http://dgidb.org/) and the GSE85871 dataset. Compared with similar publications, this study involved over 10,000 samples and more comprehensive analysis methods. Therefore, we may get more reliable and accurate information about breast cancer through this exploration. The workflow was explained in Fig. 1, and the details were provided in the Materials and Methods section.

Figure 1 Process of the present study.

GEO: Gene Expression Omnibus; TCGA: The Cancer Genome Atlas; Overlapping DEGs: Overlapping differentially expressed genes; GO: Gene Ontology; KEGG: Kyoto Encyclopedia of Genes and Genomes; PPI: protein-protein interaction; TFs, transcription factors.

Materials & Methods

Download of datasets and identification of DEGs

Five independent gene expression profiles (GSE3744, GSE21422, GSE42568, GSE61304, and GSE65194) based on GPL570 Platform (Affymetrix Human Genome U133 Plus 2.0 Array) were downloaded from GEO database to identify DEGs. After normalization between arrays, we investigated DEGs among each dataset with the threshold of |log2FoldChange (log2FC)|> 1 and adj.P.Val < 0.05 using LIMMA package of R (Ritchie et al., 2015).

For validation, GEPIA2 online tool (http://gepia2.cancer-pku.cn/#index) was used to analyze the differential expression of TCGA Breast invasive carcinoma (BRCA) RNA-seq dataset composed of 1,085 tumor samples and 112 normal samples according to the cut-off standard (|log2FC|>1 and q-value <0.05). Lastly, the common results of TCGA BRCA dataset and GEO datasets were selected as the overlapping DEGs of breast cancer, which could reduce the influences resulted from the heterogeneity of the different datasets. Venn diagram and volcano plot were drawn by VennDiagram and other packages of R. Table 1 listed the details of datasets.

Table 1 Characteristics of datasets in this study.

Expression profile dataset	Breast cancer	Normal	
GSE3744	40	7	
GSE21422	14	5	
GSE42568	104	17	
GSE61304	58	4	
GSE65194	167	11	
TCGA BRCA	1085	112	

Protein–protein interaction (PPI) network integration and hub genes screening

STRING (version 11.0; https://string-db.org/) is a biological database designed to analyze functional interactions between proteins (Szklarczyk et al., 2019). In this study, we used STRING to construct a PPI network with overlapping DEGs under the premise of an Interaction score of 0.7. Then, we utilized the cytoHubba plug-in of Cytoscape (version 3.7.2) which provided the calculated results by maximal clique centrality (MCC), maximum neighborhood component (MNC) and Degree methods to identify hub genes from the PPI network (Chin et al., 2014).

Verification of hub genes

The Oncomine database (https://www.oncomine.org) was used to verify the mRNA expression of hub genes with the threshold of P <0.05 and fold change >2. Next, the Human Protein Atlas database (HPA, https://www.proteinatlas.org) was used to validate the protein expression of genes by immunohistochemistry data.

BC-GenExMiner (http://bcgenex.centregauducheau.fr/BC-GEM/GEM-Accueil.php) is a statistical mining tool that contains published breast cancer transcription data (10,716 DNA microarray samples and 4,712 RNA-seq samples) (Jézéquel et al., 2012). We performed correlation analysis between hub genes in breast cancer using BC-GenExMiner and GEPIA2 online tool.

GO functional and KEGG pathway enrichment analysis

ClusterProfiler package of R can automatically classify biological terms and gene clusters (Yu et al., 2012). To elucidate the biological characteristics of breast cancer-related genes, we performed GO functional and KEGG pathway enrichment analysis by ClusterProfiler with p-value <0.05 and q-value <0.05.

Analysis of genetic alterations of hub genes

The cBioPortal for Cancer Genomics (cBioPortal; http://cbioportal.org) provides online resources for the exploration, visualization and analysis of multidimensional cancer genomics data (Gao et al., 2013). In this study, 6618 breast cancer samples from 13 related reports in cBioPortal were used as research materials to explore genetic alterations connected with the selected hub genes. Afterward, we utilized COSMIC (https://cancer.sanger.ac.uk/cosmic), the most comprehensive resource for studying somatic mutation information in human cancer, to analyze hub genes alterations in breast cancer (Forbes et al., 2010).

Evaluation of clinicopathological characteristics and immune infiltration

We used BC-GenExMiner to analyze the correlations between hub genes expression and clinicopathological variables such as Oestrogen receptor status (ER), Progesterone receptor status (PR), HER2 receptor status (HER2), Nodal status (N), Scarff Bloom & Richardson grade status (SBR), Nottingham Prognostic Index status (NPI), Age status, P53 status, Basal-like and Triple negative breast cancer (TNBC) subtypes. P < 0.05 was considered to be statistically significant.

TIMER (https://cistrome.shinyapps.io/timer/) is a comprehensive resource for systematic analysis of tumor-infiltrating immune cells across 32 different cancers from TCGA database (Li et al., 2017). In this experiment, we estimated the associations between hub genes expression and immune cell populations (B Cell, CD8+ T Cell, CD4+ T Cell, Macrophage, Neutrophil, and Dendritic Cell) in breast cancer using TIMER.

Prediction of TFs-hub genes and miRNAs-hub genes interaction

TRRUST (version 2, https://www.grnpedia.org/trrust/), a manually curated database of human and mouse transcriptional regulatory networks, was used to explore the TFs of hub genes (Han et al., 2018). Meanwhile, we used Encyclopedia of RNA Interactomes Platform (ENCORI, http://starbase.sysu.edu.cn/) to unearth the miRNAs targeting hub genes (Li et al., 2014). In addition, we utilized GEO2R online tool (https://www.ncbi.nlm.nih.gov/geo/geo2r/) to analyze breast cancer differentially expressed miRNAs (DEmiRNAs) in GSE97811 dataset including 45 tumor tissues and 16 normal tissues. Then, we got miRNAs-hub genes interaction combined with miRNAs-target genes analysis and DEmiRNAs data. Finally, Cytoscape was used to visualize transcriptional and post-transcriptional regulatory networks.

Survival analysis

In this section, we applied the Kaplan–Meier Plotter (http://kmplot.com/analysis/) to evaluate prognostic information of previously identified hub genes and important reporter regulatory molecules (Nagy et al., 2018). The expression values of these genes were split into either high (expression value ≥ median) or low (expression value <median). Hazard ratio (HR) was calculated to evaluate the association between genes expression and survival, and p < 0.05 was considered statistically significant.

Small molecule drugs analysis

The DGIdb online tool (http://www.dgidb.org/)—an available resource containing drug-gene interaction information from more than 30 databases—was used to screen antineoplastic drugs targeting hub genes (Cotto et al., 2018). We also downloaded the GSE85871 dataset, which is a gene expression data of MCF7 cells treated with 102 traditional Chinese medicine (TCM) ingredients recorded in the GEO database. LIMMA package of R was used to analyze the differential expression genes in each TCM ingredient treatment group compared with the untreated group (adj .p. val <  0.05). Then, we evaluated the reversal effects of each TCM ingredient on overlapping DEGs induced by breast cancer.

Results

Dataset processing and DEGs acquisition

After data normalization, the black lines of gene expression box plots of all samples in each single dataset were almost on the same level, which was an important marker to predict the reliability and accuracy of the experimental results (Fig. S1). Next, DEGs (1,738 in GSE3744, 2,430 in GSE21422, 3,116 in GSE42568, 990 in GSE61304 and 4,181 in GSE65194) were identified (Figs. 2A–2E; Tables S1–S5), and 302 integrated DEGs (110 up- and 192 down-regulated) from 5 GEO datasets were found (Figs. 2F and 2G). Similarly, we identified 3,559 DEGs from TCGA BRCA dataset with the cut-off criteria of |Log2FC| > 1 and q-value < 0.05 (Table S6). Then, 283 overlapping DEGs which might play promoting or inhibitory roles in breast cancer progression were confirmed from the analysis results of GEO datasets and TCGA BRCA dataset (Figs. 2H and 2I).

Figure 2 Identification of differentially expressed genes (DEGs) between breast cancer tumor tissues and normal tissues.

(A–E) Volcano plots of differential expression analysis for GSE3744, GSE21422, GSE42568, GSE61304, and GSE65194. Red: up-regulated; green: down-regulated. (F) Up-regulated DEGs selected by integrating five GEO datasets. (G) Down-regulated DEGs selected by integrating 5 GEO datasets. (H) Up-regulated overlapping DEGs obtained by integrating GEO datasets and TCGA BRCA dataset. (I) Down-regulated overlapping DEGs obtained by integrating GEO datasets and TCGA BRCA dataset.

PPI network construction and hub genes filtering

The PPI network around proteins encoded by 283 overlapping DEGs was constructed using STRING (Fig. S2A, Table S7). We found that 165 of the 283 overlapping DEGs were related to each other and were visualized using Cytoscape –165 nodes and 1,861 edges were included in PPI network and six hub genes (RRM2, CDC20, CCNB2, BUB1B, CDK1, and CCNA2) based on MCC, MNC and Degree methods were identified (Fig. S2B). Notably, these hub genes were all up-regulated in overlapping DEGs (Tables S1–S6) and might play important roles in the pathogenesis of breast cancer.

Verification of hub genes

Based on the large-scale breast cancer-related data in the Oncomine database, we confirmed that hub genes were significantly up-regulated in multiple cancer types, including Breast Cancer, Brain and CNS Cancer, Lymphoma, Lung Cancer, and so on (Fig. 3A). Also, immunohistochemistry staining data obtained from the HPA database demonstrated the up-regulated expression of proteins encoded by RRM2, CDC20, CCNB2, CDK1 and CCNA2 (Figs. 3B–3K). However, we did not find the association between BUB1B and breast cancer in HPA database. According to the current analysis, we predicted that BUB1B might also be associated with breast cancer, but experimental data were needed to confirm this specific connection. Meanwhile, CDK1, CCNA2, CCNB2, BUB1B and CDC20 were classified as cancer-related genes, and RRM2 was an FDA approved drug target. The data of BC-GenExMiner and GEPIA2 both confirmed a powerful correlation among hub genes (Fig. S3), suggesting that these genes might be the functional partners in breast cancer.

Figure 3 Expression verification of six hub genes.

(A) Hub genes expression in multiple types of human cancers from Oncomine database. Red: up-regulated; blue: down-regulated. (B–K) Immunohistochemical staining analysis of hub genes in breast cancer based on the Human Protein Atlas. (B) Protein levels of RRM2 in normal tissue (staining: not detected; intensity: negative; quantity: none). (C) Protein levels of RRM2 in tumor tissue (staining: low; intensity: moderate; quantity: < 25%). (D) Protein levels of CDC20 in normal tissue (staining: not detected; intensity: negative; quantity: none). (E) Protein levels of CDC20 in tumor tissue (staining: high; intensity: strong; quantity: 75%–25%). (F) Protein levels of CCNB2 in normal tissue (staining: not detected; intensity: negative; quantity: none). (G) Protein levels of CCNB2 in tumor tissue (staining: medium; intensity: moderate; quantity: 75%–25%). (H) Protein levels of CDK1 in normal tissue (staining: not detected; intensity: negative; quantity: none). (I) Protein levels of CDK1 in tumor tissue (staining: high; intensity: strong; quantity: 75%–25%). (J) Protein levels of CCNA2 in normal tissue (staining: not detected; intensity: negative; quantity: none). (K) Protein levels of CCNA2 in tumor tissue (staining: medium; intensity: strong; quantity: < 25%). There were no related immunohistochemical samples of BUB1B in the database.

GO function enrichment and KEGG pathway analysis

The GO function annotations of overlapping DEGs were mainly classified into biological processes (BP), cell component (CC) and molecular function (MF). As for BP, up-regulated overlapping DEGs were significantly related to mitotic nuclear division, organelle fission, and regulation of chromosome segregation, which was consistent with the biological characteristics of the abnormally rapid proliferation of breast cancer cells. And the down-regulated genes were closely related to regulation of cellular response to growth factor stimulus, temperature homeostasis, retinoid metabolic process and regulation of vasculature development. Within CC, the up-regulated genes were remarkably correlated to spindle, chromosome, centromeric region, condensed chromosome kinetochore, and midbody, whereas the down-regulated genes were related to collagen-containing extracellular matrix and sarcolemma. MF analysis displayed the up-regulated genes were involved in microtubule binding, microtubule motor activity, and cyclin-dependent protein serine/threonine kinase regulator activity, whereas the down-regulated genes were mainly enriched in glycosaminoglycan binding, heparin binding, extracellular matrix structural constituent and growth factor binding (Figs. 4A and 4B; Tables S8 and S9).

Figure 4 Gene Ontology (GO) and Kyoto Encyclopedia of Genes and Genomes (KEGG) pathway enrichment analysis of overlapping DEGs by ClusterProfiler and GOplot packages of R.

(A) GO analysis of up-regulated overlapping DEGs. (B) GO analysis of down-regulated overlapping DEGs. BP, biological process. CC, cellular component. MF, molecular function. (C) KEGG pathway enrichment analysis of overlapping DEGs.

KEGG pathway analysis showed the up- and down-regulated overlapping DEGs were all significantly attached to Cell cycle, Oocyte meiosis, Tyrosine metabolism, ECM-receptor interaction, Progesterone-mediated oocyte maturation, p53 signaling pathway, PPAR signaling pathway and Phenylalanine metabolism (Fig. 4C). Furthermore, the related pathways of hub genes included Cell cycle, Oocyte meiosis, Progesterone-mediated oocyte maturation, and p53 signaling pathway (Fig. 4C). Table S10 presented the detailed results of KEGG enrichment analysis.

Genetic alterations of hub genes

As a result, there were nearly 2.1% (CDC20, CDK1), 1.2% (RRM2), 0.9% (BUB1B), 0.8% (CCNA2), 0.7% (CCNB2) of breast cancer samples included in cBioPortal had genetic changes. Amplification was the most common genetic alterations among six hub genes. And deep deletion was another major genetic change among five hub genes (RRM2, CCNB2, BUB1B, CDK1, CCNA2). While genetic alterations in RRM2, CDC20, BUB1B, CDK1 and CCNA2 were related to missense mutation (Fig. S4A). In the query results of COSMIC database, we found that all hub genes had missense mutations. RRM2, CCNB2, CDK1, BUB1B and CDC20 experienced synonymous substitutions. CCNA2 and BUB1B also had frameshift deletions. Specifically, these hub genes had mutations such as A > G, A > T, C > T, C > G, G > A, G > T at nucleotide level (Fig. S4B).

Clinicopathological characteristics and immune infiltration

We investigated the relevance of six hub genes and clinicopathological features using BC-GenExMiner (Figs. S5–S7). Data analysis showed that higher expression of hub genes was found in higher NPI and SBR grade (p < 0.001). And the expression of these hub genes was significantly higher in ER-, PR-, HER2+, Nodal+, P53-mutated, Basal-like and TNBC clinical subtypes of breast cancer. Surprisingly, significantly increased expression of hub genes was found in patients not more than 51 years old (p < 0.001). Moreover, our current results demonstrated that hub genes were correlated with 6 types of immune cell infiltrates (B Cell, CD8+ T Cell, CD4+ T Cell, Macrophage, Neutrophil and Dendritic Cell) with various degrees based on the TIMER database (Fig. S8).

Prediction of TF-hub genes and miRNAs-hub genes interaction

We screened the potential regulatory relationships between TFs and hub genes via TRRUST database to further study the functional roles of hub genes. A total of 19 associations between 17 TFs and six hub genes were shown (Fig. 5A). E2F1 could activate the transcriptional process of RRM2 and CDK1. In contrast, TP53 inhibited the expression of CDK1 and CCNB2. Noticeably, PTTG1 gene, the transcriptional promoter of CDK1, was up-regulated in breast cancer; ZBTB16 gene was the transcriptional suppressor of CCNA2, and its expression was down-regulated in overlapping DEGs. Additionally, we obtained 273 targeted miRNAs with regulatory effects on hub genes using ENCORI online tool, and identified 103 DEmiRNAs (58 up- and 45 down-regulated; Table S11) from GSE97811 dataset with |log2FC|>1 and adj.P.Val < 0.05. As a result, 41 associations between 30 miRNAs and 4 hub genes (RRM2, CDK1, CCNA2, and BUB1B) were found from targeted miRNAs analysis and DEmiRNAs data (Fig. 5B). The top 2 hub genes with the most miRNAs targets were RRM2 and CCNA2. In addition, hsa-miR-340-5p, hsa-miR-130a-3p, hsa-miR-200b-3p, hsa-miR-200c-3p, hsa-miR-204-5p, hsa-miR-219a-5p, hsa-miR-27a-3p, hsa-miR-27b-3p, hsa-miR-301a-3p and hsa-miR-429 were the top 10 miRNAs with the most target genes. Unfortunately, we only used TRRUST and ENCORI databases, as well as limited samples of GSE97811 dataset to analyze TFs and miRNAs targeting hub genes, which may potentially limit the completeness of this study.

Figure 5 TFs-hub genes and miRNAs-hub genes regulatory networks.

(A) TFs-hub genes network obtained from TRRUST database. Red diamonds: hub genes; ellipses: TFs; red ellipses: up-regulated genes in overlapping DEGs; blue ellipses: down-regulated genes in overlapping DEGs; Delta-shaped arrows: activation of hub genes by TFs; half circular arrows: repression of hub genes by TFs. TFs, transcription factors. (B) miRNAs-hub genes network. Yellow diamonds: hub genes; red rectangles: up-regulated miRNAs; Blue rectangles: down-regulated miRNAs.

Survival analysis of hub genes

We then analyzed the prognostic information of six hub genes using Kaplan–Meier Plotter. The result demonstrated that breast cancer patients with higher hub genes expression had worse overall survival (OS), relapse-free survival (RFS) and distant metastasis-free survival (DMFS) (Figs. 6A–6R).

Figure 6 OS (A–F), RFS (G–L), and DMFS (M–R) analysis of RRM2, CDC20, CCNB2, BUB1B/SSK1, CDK1/CDC2, and CCNA2/CCNA in breast cancer based on Kaplan Meier-Plotter.

The patients were split into high and low expression groups according to the median expression of hub genes. OS, overall survival; RFS, relapse-free survival; DMFS, distant metastasis-free survival.

Furthermore, we performed prognostic analysis on the important reporter regulatory molecules and found that higher E2F1 and PTTG1 expression predicted worse OS. In contrast, higher expression of ZBTB16, hsa-miR-130a-3p and hsa-miR-204-5p was significantly associated with better OS in breast cancer (Fig. S9A). The remaining reporter regulatory molecules (e.g., TP53, hsa-miR-340-5p, hsa-miR-200b-3p, hsa-miR-200c-3p, hsa-miR-219a-5p, hsa-miR-27a-3p, hsa-miR-27b-3p, hsa-miR-301a-3p, hsa-miR-429) had no statistically significant correlation with OS (P >0.05; Fig. S9B).

Small molecule drugs analysis

Regarding six hub genes as potential therapeutic targets for breast cancer, we identified several antineoplastic drugs based on the DGIdb database. At present, only RRM2, CDK1 and CCNA2 were identified as tumor therapeutic targets. Therefore, we speculated the other three genes (CCNB2, BUB1B, CDC20) might be the novel targets in the future. Statistical analysis revealed that 21 candidate drugs such as Cladribine, Gallium nitrate, Dinaciclib, Alvocidib and Suramin targeted RRM2, CDK1 and CCNA2 (Table 2). More experimental data are needed to further confirm the potential of these drug candidates in the treatment of breast cancer.

Table 2 Antineoplastic drugs targeting hub genes based on the DGIdb database.

Target	Drug	Type	Sources	PMIDs	Score	
RRM2	CLADRIBINE	inhibitor	DrugBank	17852710
16316309
19576186
9923554
19715446	6	
GALLIUM NITRATE	inhibitor	ChemblInteractions DrugBank	12776257
1335254
15651176	5	
MOTEXAFIN GADOLINIUM	inhibitor	TdgClinicalTrial DrugBank TTD	–	3	
HYDROXYUREA	inhibitor	GuideToPharmacologyInteractions ChemblInteractions	–	2	
CLOFARABINE	inhibitor	GuideToPharmacologyInteractions ChemblInteractions	–	2	
GEMCITABINE	inhibitor	ClearityFoundationClinicalTrial GuideToPharmacologyInteractions	–	2	
FLUDARABINE PHOSPHATE	inhibitor	ChemblInteractions	–	1	
TRIAPINE	–	TdgClinicalTrial	–	1	
FLUDARABINE	inhibitor	GuideToPharmacologyInteractions	–	1	
CDK1	DINACICLIB	inhibitor	MyCancerGenome ClearityFoundationClinicalTrial GuideToPharmacologyInteractions ChemblInteractions CancerCommons MyCancerGenomeClinicalTrial	–	6	
ALVOCIDIB	inhibitor	MyCancerGenome TdgClinicalTrial ChemblInteractions DrugBank	11752352	5	
Roniciclib	inhibitor	GuideToPharmacologyInteractions ChemblInteractions CancerCommons	–	3	
AT-7519	inhibitor	GuideToPharmacologyInteractions ChemblInteractions DrugBank	–	3	
AZD-5438	inhibitor	GuideToPharmacologyInteractions ChemblInteractions	–	2	
TG-02	inhibitor	GuideToPharmacologyInteractions ChemblInteractions	–	2	
CHEMBL1236539	inhibitor	GuideToPharmacologyInteractions	–	1	
RG-547	inhibitor	ChemblInteractions	–	1	
SELICICLIB	inhibitor	ChemblInteractions	–	1	
CCNA2	SURAMIN	–	NCI	10208280	2	
CORDYCEPIN	–	NCI	11566717	2	
GENISTEIN	–	NCI	9664138	2	

In addition, we identified 8 TCM ingredients (1β-hydroxyalantolactone, Andrographolide, Berberine hydrochloride, Britanin, Hyodeoxycholic acid, Japonicone A, Nitidine chloride and Tanshinone IIA) that reversed breast cancer-induced gene expression from GSE85871 dataset. Japonicone A reversed the expression of 87 overlapping DEGs, including six hub genes, and its potential therapeutic effects on breast cancer were related to Cell cycle, Oocyte meiosis, p53 signaling pathway, Progesterone-mediated oocyte maturation, Viral carcinogenesis, HTLV-I infection, Pathways in cancer and TGF-beta signaling pathway (Fig. 7A). Also, Nitidine chloride, Berberine hydrochloride, 1β-hydroxyalantolactone, Britanin and Tanshinone IIA reversed the expression of 37, 35, 31, 30 and 19 overlapping DEGs, respectively, and the potential therapeutic effects of these ingredients on breast cancer were related to biological pathways such as Cell cycle, Oocyte meiosis, p53 signaling pathway and Progesterone-mediated oocyte maturation (Figs. 7B–7F). Details of TCM ingredients were shown in Table S12.

Figure 7 The reversal effects of traditional Chinese medicine (TCM) ingredients on overlapping DEGs induced by breast cancer

(A) Japonicone A. (B) Nitidine chloride. (C) Berberine hydrochloride. (D) 1β-hydroxyalantolactone. (E) Britanin. (F) Tanshinone IIA. Yellow triangle: TCM ingredient; blue diamond: down-regulated gene; Red ellipse: up-regulated gene; green rectangle: KEGG pathway.

Discussion

Although the detection and treatment of breast cancer have improved, it is still one of the most prevalent malignant tumors with the highest increase in prevalence among women (Ghoncheh, Pournamdar & Salehiniya, 2016). The diagnosis, treatment and prognosis of breast cancer have always been concerned with the world. Gene expression profiles are widely used to explore the molecular mechanisms related to tumorigenesis, which have provided valuable reference and information for clinical applications (Mohr et al., 2002).

In this study, we identified 283 overlapping DEGs (105 up- and 178 down-regulated) and six hub genes (RRM2, CDC20, CCNB2, BUB1B, CDK1, CCNA2) associated with breast cancer tumorigenesis and progression based on multiple datasets. By integrating the Oncomine, HPA, GEPIA2 and BC-GenExMiner databases, we confirmed that hub genes were over-expressed at mRNA and protein levels in breast cancer tissues compared with normal and non-cancerous tissues, and there was a powerful correlation between these genes, suggesting that hub genes were potential functional partners closely related to breast cancer. Furthermore, higher expression of hub genes was found in ER-, PR-, HER2+, Nodal+, Basal-like, P53-mutated and TNBC clinical subtypes of breast cancer, and there was a higher hub genes expression in patients not more than 51 years old. We also discovered that the over-expression of each hub gene was associated with poor OS, RFS and DMFS among patients with breast cancer, suggesting that these genes might be potential prognostic biomarkers and promote the progression of breast cancer. Further, we found the expression of hub genes was significantly correlated with immune cell infiltrates and purity, implicating that these hub genes played important roles in manipulating breast cancer immune microenvironment.

KEGG enrichment analysis showed that overlapping DEGs including 6 hub genes were significantly associated with Cell cycle and Oocyte meiosis biological pathways, and overlapping DEGs were also involved in Tyrosine metabolism, ECM-receptor interaction, Progesterone-mediated oocyte maturation and PPAR signaling pathways. To our knowledge, abnormal regulations of cell cycle and cell growth were the major causes of tumorigenesis (Zhuang et al., 2015). In the current study, we found that two important pathways related to cell growth and apoptosis—Cell cycle and Oocyte meiosis—were dysregulated in breast cancer. In addition, ECM-receptor interaction pathway played important roles in tumor shedding, adhesion, degradation, movement and hyperplasia. So far, some studies have shown that ECM receptor interaction pathway was closely related to breast cancer, colorectal cancer, head and neck cancer and other human tumors (Bao et al., 2019; Islam et al., 2018; Rahman et al., 2019). Gasco, Shami & Crook (2002) concluded that molecular pathological analysis of specific components of p53 signaling pathway may be helpful for the diagnosis and prognosis of breast cancer. In addition, it has been found that signal transduction pathways such as Tyrosine metabolism, Progesterone-mediated oocyte maturation and PPAR signaling pathway may also be associated with the occurrence of human cancers (Chen et al., 2012; Liu & Ye, 2017; Pietras et al., 1995). These pathways provided insights into the molecular mechanisms of breast cancer initiation and development.

Accumulating studies have demonstrated that CDK1, CCNB2, CCNA2, CDC20 and BUB1B, as genes related to cell cycle, are involved in the occurrence and development of tumors. CDK1, also known as CDC2, plays an important role in the precise cell division (Kang et al., 2014). Inhibiting the expression of CDK1 can suppress tumor cells growth and induce apoptosis in TNBC clinical subtype of breast cancer (Liu et al., 2014). In addition, high expression of CDK1 led to worse 5-year RFS in breast cancer patients (Kim et al., 2008), similar to the experimental results (Fig. 6K). As an important component in cell cycle regulation, CCNB2 seems to functions as the oncogene and independent prognostic factor for survival in patients with breast cancer (Shubbar et al. 2013). Tang et al. (2018) proved that there was a significant correlation between CCNB2 and molecular subtypes of breast cancer (Fig. S6A). CCNA2, a key regulator of cell cycle, could promote the transformation and progression of cancer (He et al., 2017). Gao et al. (2014) found the over-expression of CCNA2 in breast cancer was related to the unfavorable prognosis , similar to our current findings (Fig. 6). We understand that there have been previous reports that have associated CDC20 over-expression with tumor progression and poor prognosis of breast cancer, indicating that CDC20 may be a useful marker for monitoring breast cancer progression (Sewart & Hauf, 2017). Similarly, BUB1B played a pivotal role in the proliferation and progression of many tumors (Takagi et al., 2013). RRM2, a breast cancer hub gene participated in phenylalanine metabolic pathway in our study, was closely linked to tumor growth, invasion, angiogenesis, tumor metastasis and other cellular functions, as well as the prognosis of breast cancer patients (Bell, Barraclough & Vasieva, 2017; Chen et al., 2019). It has been previously reported that RRM2 expression was associated with the resistance of tumorigenic breast cancer cells to chemotherapy (Shah et al., 2015). Taken together, our findings were consistent with other previous studies that six hub genes may serve as predictive biomarkers for diagnosis and prognosis of patients with breast cancer.

We also found that several reporter regulatory molecules (e.g., E2F1, TP53, PTTG1, ZBTB16, hsa-miR-340-5p, hsa-miR-130a-3p, hsa-miR-200b-3p, hsa-miR-204-5p) regulated the transcription or post-transcription of hub genes associated with major biological processes and pathways in breast cancer, and they were also related to the OS of breast cancer patients. E2F1 and ZBTB16 have been proved to belong to tumor-suppressive genes (Wasim et al., 2010; Worku et al., 2008). In contrast, PTTG1 and mutated TP53 promoted the proliferation of tumor cells (Fu, Zhang & Cui, 2018; Gasco, Shami & Crook, 2002).

Breast cancer also shows changes in the expressions of noncoding RNAs (e.g., miRNAs, additional elements). Several studies have previously proven that miRNAs played vital roles in many biological processes, including cell growth, differentiation, metabolism, apoptosis and signal transduction. For instance, (Ma (2019) demonstrated that miR-219-5p inhibited the cell proliferation and cell cycle distribution of ESCC cells by inhibiting the expression of CCNA2, highlighting the role of miR-219-5p and CCNA2 in cell cycle and tumor growth. Liang et al. (2019) confirmed the over-expression of miR-204-5p not only inhibited the high expression of RRM2 in breast cancer cells but also inhibited cell migration and invasion of breast cancer. The above conclusions further demonstrated the miRNAs identified in this study were promising biomarkers for breast cancer. Recent publications described that additional elements including REP522, D20S16, HERVKC4-INT and HERV1_LTRc were also abnormally expressed in ER+/HER2- breast cancer, showing that it is necessary to further strengthen the study of these additional elements related to cancer (Karakülah et al., 2019; Yandım & Karakülah, 2019).

Next, we further identified 21 anti-tumor drugs (e.g., Cladribine, Gallium nitrate, Dinaciclib, Alvocidib, Suramin) targeting RRM2, CDK1 and CCNA2 (Table 2). Nevertheless, whether these drugs could exert therapeutic effects on breast cancer by inhibiting the over-expression of RRM2, CDK1 and CCNA2, or whether CCNB2, BUB1B and CDC20 are promising therapeutic targets still need to be supported by further research. In addition, we identified 8 TCM ingredients (1β-hydroxyalantolactone, Andrographolide, Berberine hydrochloride, Britanin, Hyodeoxycholic acid, Japonicone A, Nitidine chloride and Tanshinone IIA) that reversed breast cancer-induced overlapping DEGs expression. Previous limited studies have reported that some of TCM ingredients mentioned above affected the cell cycle of tumor cells (Du et al., 2015; Li et al., 2020a; Lu, 2009; Pan et al., 2011; Wang et al., 2016; Zou et al., 2017), similar to our current study that these ingredients inhibited proliferation as well as promoted apoptosis of breast cancer cells through more than one biological pathway (Fig. 7).

Conclusions

In the current study, we obtained 283 overlapping DEGs and six hub genes (RRM2, CDC20, CCNB2, BUB1B, CDK1, and CCNA2) related to the occurrence and development of breast cancer via comprehensive bioinformatics analysis. Furthermore, we considered the associations between hub genes expression and clinicopathological factors (e.g., age, subtypes) of patients with breast cancer. The study also established TFs-hub genes and miRNAs-hub genes networks, and found several reporter regulatory molecules (e.g., E2F1, PTTG1, TP53, ZBTB16, hsa-miR-130a-3p, hsa-miR-204-5p) significantly related to the progression and prognosis of breast cancer. Meanwhile, 29 small molecule drugs with potential therapeutic effects for breast cancer were identified from the DGIdb database and GSE85871 dataset. In summary, our research provided further clues for breast cancer therapeutic drugs and biomarkers.

Supplemental Information

Supplemental Information 1 Normalization of datasets

(A) Normalization of GSE3744. (B) Normalization of GSE21422. (C) Normalization of GSE42568. (D) Normalization of GSE61304 . (E) Normalization of GSE65194. Vertical axis: expression value; Horizontal axis: sample list. Black: normal; Red: tumor.

Click here for additional data file.

Supplemental Information 2 Protein-protein interaction (PPI) network around proteins encoded by overlapping DEGs

(A) A PPI network was constructed via STRING. (B) six hub genes were identified by MCC, MNC and Degree methods.

Click here for additional data file.

Supplemental Information 3 Correlation analysis between hub genes in breast cancer

(A) Expression correlation analysis of six hub genes based on GEPIA2 database. (B) Expression correlation verification of six hub genes in all DNA microarray data and RNA-seq data related to breast cancer based on BC-GenExMiner tool.

Click here for additional data file.

Supplemental Information 4 Analysis of genetic alterations of six hub genes in breast cancer based on cBioPortal (A) and COSMIC (B) databases, respectively

Click here for additional data file.

Supplemental Information 5 RRM2 and CDC20 expression patterns by NPI, SBR, ER status, PR status, HER2 status, Nodal status, Basal-like status, TNBC status, P53-mutated status and age group

Click here for additional data file.

Supplemental Information 6 CCNB2 and BUB1B expression patterns by NPI, SBR, ER status, PR status, HER2 status, Nodal status, Basal-like status, TNBC status, P53-mutated status and age group

Click here for additional data file.

Supplemental Information 7 CDK1 and CCNA2 expression patterns by NPI, SBR, ER status, PR status, HER2 status, Nodal status, Basal-like status, TNBC status, P53-mutated status and age group

Click here for additional data file.

Supplemental Information 8 Correlations between six hub genes expression and immune infiltrates in breast cancer

(A) RRM2. (B) CDC20. (C) CCNB2. (D) BUB1B. (E) CDK1. (F) CCNA2.

Click here for additional data file.

Supplemental Information 9 OS analysis of the important reporter regulatory molecules in breast cancer

(A) Reporter regulatory factors that are significantly related to OS. (B) Some reporter regulatory factors that have no statistically significant correlation with OS.

Click here for additional data file.

Supplemental Information 10 Differentially expressed genes (DEGs) in dataset GSE3744

Click here for additional data file.

Supplemental Information 11 Differentially expressed genes (DEGs) in dataset GSE21422

Click here for additional data file.

Supplemental Information 12 Differentially expressed genes (DEGs) in dataset GSE42568

Click here for additional data file.

Supplemental Information 13 Differentially expressed genes (DEGs) in dataset GSE61304

Click here for additional data file.

Supplemental Information 14 Differentially expressed genes (DEGs) in dataset GSE65194

Click here for additional data file.

Supplemental Information 15 Differentially expressed genes (DEGs) in TCGA BRCA dataset

Click here for additional data file.

Supplemental Information 16 PPI network around proteins encoded by overlapping DEGs

Click here for additional data file.

Supplemental Information 17 GO functional enrichment analysis of up-regulated overlapping DEGs

Click here for additional data file.

Supplemental Information 18 GO functional enrichment analysis of down-regulated overlapping DEGs

Click here for additional data file.

Supplemental Information 19 KEGG pathway enrichment analysis of overlapping DEGs

Click here for additional data file.

Supplemental Information 20 Differentially expressed miRNAs (DEmiRNAs) in dataset GSE97811

Click here for additional data file.

Supplemental Information 21 Traditional Chinese medicine (TCM) ingredients targeting overlapping DEGs

Click here for additional data file.

Abbreviations

DEGs differentially expressed genes

GEO Gene Expression Omnibus

TCGA The Cancer Genome Atlas

PPI protein–protein interaction

GO Gene Ontology

KEGG Kyoto Encyclopedia of Genes and Genomes

TFs transcription factors

TCM traditional Chinese medicine

Log2FC log2FoldChange

BRCA Breast invasive carcinoma

MCC maximal clique centrality

MNC maximum neighbourhood component

HPA Human Protein Atlas

ER Oestrogen receptor status

PR Progesterone receptor status

HER2 HER2 receptor status

SBR Scarff Bloom & Richardson grade status

NPI Nottingham Prognostic Index status

TNBC Triple negative breast cancer

ENCORI Encyclopedia of RNA Interactomes platform

DEmiRNAs differentially expressed miRNAs

HR hazard ratio

OS overall survival

RFS relapse-free survival

DMFS distant metastasis-free survival

Additional Information and Declarations

Competing Interests

Author Contributions

Data Availability

The authors declare there are no competing interests.

Mingqian Hao conceived and designed the experiments, performed the experiments, prepared figures and/or tables, authored or reviewed drafts of the paper, and approved the final draft.

Wencong Liu conceived and designed the experiments, authored or reviewed drafts of the paper, and approved the final draft.

Chuanbo Ding conceived and designed the experiments, analyzed the data, authored or reviewed drafts of the paper, and approved the final draft.

Xiaojuan Peng, Yue Zhang, Huiying Chen and Yingchun Zhao performed the experiments, prepared figures and/or tables, and approved the final draft.

Ling Dong and Xueyan Chen analyzed the data, authored or reviewed drafts of the paper, and approved the final draft.

Xinglong Liu analyzed the data, prepared figures and/or tables, and approved the final draft.

Sadia Khatoon performed the experiments, authored or reviewed drafts of the paper, and approved the final draft.

Yinan Zheng conceived and designed the experiments, authored or reviewed drafts of the paper, and approved the final draft.

The following information was supplied regarding data availability:

The differentially expressed genes data of datasets GSE3744, GSE21422, GSE42568, GSE61304, GSE65194, TCGA BRCA and GSE97811 are available in Tables S1–S6 and Tables S11, respectively. The PPI network data in this article is available in Tables S7. The GO functional enrichment and KEGG pathway analysis data are available in Tables S8–S10. The results of the reversal effects of 8 traditional Chinese medicine ingredients on breast cancer from the GSE85871 dataset are available in Tables S12.

The raw data of gene and miRNA expression are available at NCBI GEO (GSE42568, GSE3744, GSE21422, GSE61304, GSE65194, GSE97811 and GSE85871) and GEPIA 2 (http://gepia2.cancer-pku.cn/#degenes).

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
