# Peer review of "Identification of hub genes and small molecule therapeutic drugs related to breast cancer with comprehensive bioinformatics analysis"

_PeerJ, doi:10.7717/peerj.9946_

## Round 0.1 · original submission · Major Revisions

For further consideration of the manuscript:

- the authors satisfactorily implemented all referee comments.
- I found the intro section to be very weak and it needs to be improved substantially.
- as suggested by reviewer #3, the manuscript needs to be edited by a native-equivalent speaker to increase the readability
- there are tons of available RNA-seq data in public databases, the authors should consider taking advantage of these datasets to make more robust conclusions.
- breast cancer also shows changes in the expressions of noncoding RNAs, (e.g. PMID: 31536958 and PMID: 31824778) and one should also mention that protein-coding genes should not be the sole drivers. The authors include these in the revised version of the discussion.

Reviewer 1 ·

Basic reporting

Hao et al in this manuscript employed already established bioinformatics approaches to detect candidate biomarkers, pathways and candidate drugs of breast cancer. The results are well presented with figures and Tables.

Experimental design

Authors analyzed 5 breast cancer data sets downloaded from the GEO database.
A series of bioinformatics approach was used to evaluate the prognostic biomarkers and candidate drugs. Methods are well explained.

Validity of the findings

no comment'

Additional comments

1) The novelty of the study is not clear; can you please write how your approach is is different than others and what is the contribution of your work?
2) Please clarify in the introduction how is your work than previous works.
3) The roles of hub genes can be explained more detail in the discussion.
4) In line 343, authors claimed the identified pathways provide more insights into the development of breast cancer. This needs relevant mechanistic discussion, how the findings may improve the pathogenesis.
5) Please consult these two latest papers to clear your biomarkers and drug repositioning part of your manuscript which still needs development; best systems biology papers:
https://doi.org/10.3390/medicina55010020
https://doi.org/10.1089/omi.2018.0048

Reviewer 2 ·

Basic reporting

no comment

Experimental design

no comment

Validity of the findings

no comment

Additional comments

In general, this study was comprehensive and analysized five breast cancer data sets. Several minor concerns are as followed. First, the GEO data set selected by the author is much larger than the normal samples. How did the author avoid this statistical deviation? Second, GEO has amounts of data on breast cancer, and we recommended the authors select more sets of data for expression and survival analysis validation. Third, HUB genes screened by the author have been reported for many times, and we believe novel genes shall be explored to improve the quality of this study.

·

Basic reporting

Table 2: The borders are not added in the table and it is not clear which drugs are relating to the target gene(s).

The authors will have to work more on their introduction/background section, by adding more relevant content and doing an in-depth literature review. The fact that the abstract to the paper (26 lines) is longer than the introduction (25 lines) is an indication of how short it is!

Line 81: Perhaps the authors can use the term overlapping DEGs to refer to the genes that are differentially expressed across all the 5 datasets (also in other parts of the manuscript).

Figure 2A: There are scales present for 2 datasets, while none for others (or isn't visible clearly). Figure 2F has only 2 components, the third is not represented - were no hub genes found in this feature?

Line 187: Since the 6 hub genes were taken from the list of overlapping genes, they will be sig. dysregulated.

Most figure legends do not explain the key used and provide very little information. The figures also feel cramped, and already 10 images with 5 suppl. images. The authors can choose to move more data to the supplementary and plan their paper with 5-6 main figures.

Experimental design

One of the major (potential) pitfalls of this study is the use of 5 GEO datasets sequenced on the Affymetrix Human Genome U133 Plus 2.0 Array platform. While it is acceptable that this platform has previously been widely used, in today's context of high-throughput, deep NGS sequencing, and with the availability of many datasets from RNA-seq projects, it seems a bit outdated to use Chip data, the most latest of which is 5 years old. Perhaps the authors can throw more light on their choice of platform?

Line 94-96: Was the selection of the verification dataset done at random, or was there any reason behind the choice?

Fig. 1 can be amended to include information about the downloaded MCF-7 dataset.

Validity of the findings

The authors have used data from multiple datasets available publicly, to identify a set of genes, which were further compared against public databases/tools to assess their role as a potential target using available drugs. However, the manuscript has not been written in a concise and clear manner, thereby casting apprehensions over the validity of these results at many junctures. The work would greatly benefit from being re-written with a more simplified approach.

Additional comments

This work by Hao and colleagues titled "Identification of hub genes and screening of small molecule therapeutic drugs for breast cancer based on comprehensive bioinformatics analysis" explores published datasets for genes that can be used as targets for drugs and other sources from traditional Chinese medicine. The manuscript has many errors in different sections and would greatly benefit from being edited by a native speaker or by a professional language editor.

---

## Round 0.2 · accepted · Accept

The authors successfully implemented the criticisms and comments raised by the reviewers during the revision period. Therefore, the manuscript can be accepted in its current form.

Reviewer 1 ·

Basic reporting

no comment

Experimental design

no comment

Validity of the findings

no comment

Additional comments

The authors have addressed this reviewers comments and I would recommend to accept it.